# Pharmacologic Targeting of MMP2/9 Decreases Peritoneal Metastasis Formation of Colorectal Cancer in a Human Ex Vivo Peritoneum Culture Model

**DOI:** 10.3390/cancers14153760

**Published:** 2022-08-02

**Authors:** Jana Koch, Dina Mönch, Annika Maaß, Alina Mangold, Miodrag Gužvić, Thomas Mürdter, Tobias Leibold, Marc-H. Dahlke, Philipp Renner

**Affiliations:** 1Dr. Margarete Fischer-Bosch Institute of Clinical Pharmacology, 70376 Stuttgart, Germany; jana.koch@ikp-stuttgart.de (J.K.); dina.moench@ikp-stuttgart.de (D.M.); annika.maass@ikp-stuttgart.de (A.M.); thomas.muerdter@ikp-stuttgart.de (T.M.); 2University of Tübingen, 72074 Tübingen, Germany; 3Robert Bosch Centre for Tumour Diseases (RBCT), Department of General and Visceral Surgery, Robert Bosch Hospital, 70376 Stuttgart, Germany; alina.mangold@rbk.de (A.M.); tobias.leibold@rbk.de (T.L.); marc.dahlke@rbk.de (M.-H.D.); 4University of Regensburg, 93059 Regensburg, Germany; miodrag.guzvic@klinik.uni-regensburg.de; 5University Medical Centre Regensburg, 93053 Regensburg, Germany

**Keywords:** matrix metalloproteinases, peritoneal carcinosis, ex vivo model, colorectal cancer, small molecule inhibitors

## Abstract

**Simple Summary:**

We investigated the effects of matrix metalloproteinases (MMPs) on the peritoneal attachment of colorectal cancer cells in patient samples and in a human ex vivo peritoneum model. MMP2/9 overexpression and enhanced fibronectin cleavage occurred during peritoneal colonisation, which could be inhibited by specific MMP inhibition, thereby reducing cancer cell attachment.

**Abstract:**

Background: Matrix metalloproteinases (MMPs) play a crucial role in tumour initiation, progression, and metastasis, including peritoneal carcinosis (PC) formation. MMPs serve as biomarkers for tumour progression in colorectal cancer (CRC), and MMP overexpression is associated with advanced-stage metastasis and poor survival. However, the molecular mechanisms of PC from CRC remain largely unclear. Methods: We investigated the role of MMPs during peritoneal colonisation by CRC cell lines in a human ex vivo peritoneum model and in patient-derived CRC and corresponding PC samples. MMP2 and MMP9 were inhibited using the small-molecule inhibitors batimastat and the specific MMP2/9 inhibitor III. Results: MMP2 and MMP9 were strongly upregulated in patient-derived samples and following peritoneal colonisation by CRC cells in the ex vivo model. MMP inhibition with batimastat reduced colonisation of HT29 and Colo205 cells by 36% and 68%, respectively (*p* = 0.0073 and *p* = 0.0002), while MMP2/9 inhibitor III reduced colonisation by 50% and 41%, respectively (*p* = 0.0003 and *p* = 0.0051). Fibronectin cleavage was enhanced in patient-derived samples of PC and during peritoneal colonisation in the ex vivo model, and this was inhibited by MMP2/9 inhibition. Conclusion: MMPs were upregulated in patient-derived samples and during peritoneal attachment of CRC cell lines in our ex vivo model. MMP2/9 inhibition prevented fibronectin cleavage and peritoneal colonisation by CRC cells. MMP inhibitors might thus offer a potential treatment strategy for patients with PC.

## 1. Introduction

Peritoneal carcinosis (PC) usually presents in patients with advanced stage malignancies, and shows limited response to systemic chemotherapy [1]. Cytoreductive surgery combined with hyperthermic intraperitoneal chemotherapy has been used as a potentially curative treatment approach, with a significant impact on the course of disease in otherwise palliative patients. Several, mostly retrospective, trials have demonstrated the efficacy of this therapeutic concept [2,3]. However, recent prospective trials have shown conflicting evidence regarding the benefit of HIPEC [4]. The PRODIGE 7 trial failed to show any effect of oxaliplatin-based hyperthermic intraperitoneal chemotherapy on survival in patients with PC of colorectal cancer (CRC) [5]. COLOPEC and PROPHYLOCHIP have shown that adjuvant HIPEC or second-look surgery plus HIPEC did not prevent peritoneal metastasis [6,7]. This emphasises the need for alternative therapeutic strategies, including the replacement of established agents such as mitomycin C and oxaliplatin.

PC formation is a multi-step process regulated by a complex interplay between tumour cells and various components of the peritoneum. Tumour cell adhesion to the peritoneum and migration into the tissue is predominantly promoted by adhesion molecules and matrix remodelling enzymes, such as matrix metalloproteinases (MMPs) [8,9]. MMP2 and MMP9 are members of the gelatinase sub-family of MMPs, with proteolytic activity against other extracellular matrix (ECM) molecules [10]. MMP2 is upregulated in invasive CRC, and increased MMP2 and MMP9 expression levels were shown to correlate with worse outcomes [10,11]. However, the precise roles of these proteins in the process of local metastatic progression to peritoneal carcinomatosis remain unclear.

In this study, we investigated the effects and molecular mechanisms of MMP inhibition on metastasis formation using a human ex vivo peritoneum model to mimic peritoneal carcinomatosis from CRC. The results showed that MMPs were overexpressed during peritoneal colonisation by CRC cells in the ex vivo model and that this overexpression was prevented by pharmacological inhibition of MMP2 and MMP9, leading to a marked reduction in peritoneal seeding in our functional primary culture model.

## 2. Materials and Methods

### 2.1. Cell Culture

The Colo205/GFP CRC cell line was purchased from Cellomics Technology (SC-1278; Halethorpe, MD, USA) and the HT29/GFP-luciferase and SKOV3/GFP cell lines were obtained from Julia Beil (University Hospital Tübingen, Eberhard Karls University, Internal Medicine VIII, Tübingen, Germany). Cell lines were authenticated by short tandem repeat analysis (Eurofins, Ebersberg, Germany). The cell lines were tested for mycoplasma by polymerase chain reaction (PCR) using a Venor^®^GeM Classic kit (11-1025; Minerva Biolabs, Berlin, Germany) according to the manufacturer’s instructions before use in experiments. Colo205/GFP cells were cultivated in RPMI 1640 w/Glutamax (Thermo Fisher Scientific GmbH, Waltham, MA, USA) supplemented with 1% penicillin/streptomycin (Biochrom AG, Berlin, Germany) and 10% fetal calf serum (FCS) (Thermo Fisher Scientific GmbH, Waltham, MA, USA) in a 5% CO_2_ incubator at 37 °C. HT29/GFP-luciferase and SKOV3/GFP cells were cultivated in McCoy’s 5A Medium with L-glutamine (Pan Biotech GmbH, Aidenbach, Germany) supplemented with 1% penicillin/streptomycin and 10% FSC in a 5% CO_2_ incubator at 37 °C.

### 2.2. Peritoneal Culture

Fresh peritoneal tissue was obtained directly from the operating room, with informed consent from the patient, and transferred immediately to the laboratory in E199 medium (Biochrom AG, Berlin, Germany). The tissue was then incubated for 15 min in phosphate-buffered saline (PBS)-containing penicillin/streptomycin and amphotericin (Sigma–Aldrich Chemie, St. Louis, MO, USA). Extraperitoneal fat was removed, and 7 × 7 mm tissue pieces were inserted between two stainless steel rings and cultured, with the mesothelial cell surface directed upwards, in E199 medium containing penicillin/streptomycin, L-glutamine (Biochrom AG, Berlin, Germany), FCS (Thermo Fisher Scientific GmbH, Waltham, MA, USA), hydrocortisone (Sigma–Aldrich Chemie, St. Louis, MO, USA), fibroblast growth factor (PeproTech GmbH, Hamburg, Germany), and heparin (Biochrom AG, Berlin, Germany), as described previously [12].

### 2.3. Decellularisation of Peritoneal Tissue

For decellularisation, peritoneal tissue was incubated with buffer A (10 mM Tris, 0.1% EDTA, pH 7.8) for 18 h at 37 °C. The next day, buffer A was removed, and the tissue was washed twice with PBS, followed by incubation with 0.1% sodium dodecyl sulfate (SDS) for 24 h at 37 °C. The tissue was then washed three times with buffer B (10 mM Tris, pH 7.8) and digested using 50 U/mL DNAse (AppliChem GmbH, Darmstadt, Germany) in 20 mM Tris, 2 mM MgCl_2_, pH 7.8 for 3 h at 37 °C. Decellularisation was confirmed by haematoxylin and eosin staining.

### 2.4. Co-Culture

Peritoneal tissue samples were inserted between stainless steel rings (internal diameter 4.4 or 12 mm). For co-culture, 0.2–1.5 × 10^6^ green fluorescent protein (GFP)-expressing cancer cells were seeded onto the peritoneum, depending on the ring’s diameter, and cultured in peritoneum medium for the indicated times, as described above.

### 2.5. MMP Inhibitor Treatment

Cancer cell lines were seeded on plastic dishes or peritoneal tissue and treated with MMP-2/MMP-9 inhibitor III (444251; Merck Chemicals GmbH, Darmstadt, Germany) or batimastat (196440; Sigma–Aldrich Chemie, St. Louis, MO, USA) for the indicated times. Attached GFP-expressing cells were then counted 24 h after seeding.

### 2.6. Cell Counting

To analyse the number of GFP-expressing cancer cells attached to the peritoneum, peritoneal tissue was dislodged from the rings after 24 h of co-culture, washed carefully with PBS, and then incubated with collagenase (C0130; Sigma–Aldrich Chemie, St. Louis, MO, USA) at 2 mg/mL for 10 min, to release attached cancer cells. The release of cancer cells from the peritoneum was monitored under a fluorescence microscope, and GFP-expressing cancer cells were counted using a Neubauer chamber (MARI0640110; VWR, Radnor, PA, USA).

### 2.7. Single-Cell Isolation

GFP-expressing cancer cells that colonised the peritoneum were subjected to transcriptome analysis. The peritoneal tissue was dislodged from the rings after 24 h of co-culture, rinsed with PBS, and then incubated with collagenase (C0130; Sigma–Aldrich Chemie, St. Louis, MO, USA) at 2 mg/mL for 10 min to release attached cancer cells. The release of cancer cells from the peritoneum was monitored under a fluorescence microscope. GFP-expressing single cancer cells were then isolated under the microscope using a 20 µL pipette and stored in a 4 µL mTRAP™ Lysis buffer (29011; Active Motif, Carlsbad, CA, USA) supplemented with 0.4 µL tRNA (from *Escherichia coli* MRE 600, 10109541001; Roche Diagnostics GmbH, Basel, Switzerland) at −80 °C.

### 2.8. Whole-Transcriptome Analysis (WTA) of Single Cells

Transcriptome analysis of single cells was performed as described previously [13,14].

### 2.9. Quantitative Real-Time PCR (RT^2^ and qRT-PCR Assays)

For qRT-PCR analysis, 1 µg of total RNA was reverse transcribed using a SuperScript First-Strand Synthesis System for RT-PCR kit (Life Technologies, Carlsbad, CA, USA), and 10 ng of cDNA per sample was subjected to each qRT-PCR reaction. PrimePCR SYBR Green Assays for MMP2 (qHsaCED0042560, #100025637) and MMP9 (qHsaCID0011597, #100025637) were obtained from BioRad (Feldkirchen, Germany) and used with RT^2^ SYBR Green ROX qPCR Mastermix (Qiagen, Hilden, Germany) according to the manufacturer’s instructions. The housekeeping genes PrimePCR SYBR Green Assay ribosomal protein lateral stalk subunit P0 (RPLP0) (qHsaCED0038653, #100025637) and glyceraldehyde 3-phosphate dehydrogenase (*GAPDH*) at a final concentration of 1.25 nM (forward: GCAAATTCCATGGCACCGT; reverse: TCGCCCCACTTGATTTTGG) were used for normalisation. qRT-PCR was carried out using a 7900HT Fast Real-Time PCR System (Life Technologies, Carlsbad, CA, USA), and each sample was analysed in duplicate. Gene expression was calculated using the ∆∆Ct method: MMP2 or MMP9 expression for each sample was normalised to the mean of RPLP0 and GAPDH and calculated relative to the normal tissue or untreated control.

RT^2^ analysis was performed using a Human Tumour Metastasis (PAHS-028ZE-1, Qiagen) or Custom RT^2^ Profiler PCR Array (CLAH38721) according to the manufacturer’s protocol, with 5 ng cDNA or single-strand DNA from WTA per sample.

### 2.10. siRNA Treatment

siRNA pools for MMP2, MMP9, and control were obtained as 10 nmol stocks from siTOOLs Biotech GmbH (Planegg, Germany). Lipofectamine™ RNAiMAX Transfection Reagent (13778075) and Opti-MEM™ Reduced Serum Medium (31985062) were obtained from Thermo Fisher Scientific GmbH. Reverse transfection was performed according to the manufacturer’s protocol for 96- or 6-well plates.

### 2.11. Gel Electrophoresis and Immunoblotting

Peritoneal tissue was mixed with lysis buffer (50 mM Tris-HCl pH 7.6, 250 mM NaCl, 5 mM EDTA, 0.1% Triton x-100) containing complete protease inhibitor cocktail (4693124001) and PhosSTOP phosphatase inhibitor (4906845001) (Sigma–Aldrich Chemie GmbH, St. Louis, MO, USA) and minced using an ULTRA-TURRAX^®^ (T 10 basic, IKA, Staufen, Germany) until no large particles were left. The tissue was then transferred into Lysing Matrix D tubes (6913100, MP Biomedicals, Irvine, CA, USA) for further comminution and sonicated for 20 s in a Bioruptor UCD-200 (Diagenode, Liège, Belgium). Plastic-adherent cells were harvested and washed twice with PBS, and the pellets were resuspended in lysis buffer. Cells were broken up by two cycles of ultrasound treatment for 20 s each, and protein levels were measured using a Pierce BCA Protein Assay Kit (23227; Thermo Fisher Scientific GmbH, Waltham, MA, USA) according to the manufacturer’s instructions. Protein (15–20 µg) was separated by SDS-polyacrylamide gel electrophoresis and transferred onto nitrocellulose membranes (0.45 µm). The membranes were then probed with primary antibodies against fibronectin (P1H11, Mouse mAb, #MAB1918; R&D Systems, Minneapolis, MN, USA), MMP2 (D4M2N, Rabbit mAb, #40994; Cell Signaling Technology, Danvers, MA, USA), MMP9 (D6O3H, Rabbit mAb, #13667; Cell Signaling Technology, Danvers, MA, USA), and β-actin (AC-15, Mouse mAb, Order-no. A5441, Lot no. 029M4883V; Sigma–Aldrich Chemie GmbH, St. Louis, MO, USA) overnight at 4 °C, followed by incubation with anti-rabbit IgG-horseradish peroxidase (Order-no. 7074 S, Lot no. 28; Cell Signaling Technology, Danvers, MA, USA) or anti-mouse IgG-horseradish peroxidase (Order-no. 7076, Lot no. 33; Cell Signaling Technology, Danvers, MA, USA) for 1 h at room temperature. Antibodies were used at a dilution of 1:1000, except anti-β-actin (loading control; 1:5000) and secondary antibodies (1:5000). Signals were detected by enhanced chemiluminescence (SuperSignal West Dura Extended Duration Substrate, 34075; Thermo Fisher Scientific GmbH, Waltham, MA, USA) using a charge-coupled device camera (STELLA3200; Raytest, Straubenhardt, Germany).

### 2.12. Protein Quantification

Relative protein levels were quantified using Fiji/ImageJ with the gel-analysis tool. The amounts of fibronectin and cleaved fibronectin were calculated relative to the β-actin loading control. The ratio of cleaved fibronectin to intact fibronectin was calculated to evaluate the changes following co-culture and subsequent treatments.

### 2.13. Immunohistochemistry

Formalin-fixed, paraffin-embedded sections (4 µm) of peritoneal carcinomatosis tissue were stained with Mayer’s haematoxylin (Sigma–Aldrich Chemie, St. Louis, MO, USA) and eosin (Merck Chemicals GmbH, Darmstadt, Germany). MMP immunostaining was carried out using the following antibodies and pretreatments: MMP2 (#40994, clone D4M2N; Cell Signalling Technology, Danvers, MA, USA; 1:150, heat-induced epitope retrieval at pH 9), MMP9 (NCL-MMP9-439, clone 15W2; Novacastra via Leica Biosystems, Wetzlar, Germany; 1:50, heat-induced epitope retrieval at pH 9). Afterwards, endogenous peroxidase blocking (S2023; Agilent, Santa Clara, CA, USA) was carried out for 10 min at room temperature. Primary antibody staining was performed at 4 °C overnight followed by peroxidase/3,3′-diaminobenzidine-based detection using a Dako REAL EnVision Detection System (K7005; Agilent, Santa Clara, CA, USA).

### 2.14. Statistical Analysis

Graphs were made using GraphPad Prism 5. Control and treatment conditions were compared using two-sided Student’s *t*-tests (paired/unpaired) in Microsoft Excel 2016. All results are shown as the mean and standard error of at least three independent experiments.

## 3. Results

### 3.1. MMP Overexpression in Primary and Peritoneal Human CRC Samples

We analysed MMP expression levels in primary CRC and patient-derived PC samples. Analysis of MMP2 and MMP9 mRNA expression levels in eight primary colorectal tumours revealed a 3.6–13.7-fold increase in MMP2 (*p* = 0.0356) and a 0.5–320.7-fold increase in MMP9 (*p* = 0.5286) levels compared with healthy colon tissue (Figure 1A). In patient-derived PC samples, mRNA levels of MMP9, but not MMP2, were increased 6–10.5-fold (*p* = 0.0053) compared with adjacent healthy peritoneum (*n* = 3; Figure 1B). Immunohistochemistry staining accordingly revealed the strong expression of MMP9, but not MMP2, in PC compared with healthy tissues (Figure 1C). These data showed that MMP2 and MMP9 were overexpressed in primary CRC and in the corresponding PC.

### 3.2. Upregulation of MMPs in Colonising Peritoneal Cancer Cells in a Human Ex Vivo Model

We also analysed the expression levels of MMPs and other candidate genes relevant to the PC of CRC in HT29 and Colo205 cells in our ex vivo PC model. For this, we used the previously established human ex vivo peritoneal model, which allows for the investigation of PC formation and possible treatment options in a clinically relevant ex vivo system [15] (Figure 2A). In this model, fresh human peritoneal tissue was obtained from the surgery room and directly inserted between two stainless steel rings. To mimic PC formation and to analyse the molecular mechanisms’ underlying metastasis formation, the tissue was co-cultured with GFP-transduced CRC cells to induce PC, as described previously [15] (Figure 2B). Cells were co-cultured with cellularised or decellularised peritoneum to determine the roles of cellular and acellular ECM components, respectively, and their effects on gene expression in CRC cells during co-culture. Attached GFP-expressing cells were isolated for the analysis of gene expression during PC formation 24 h after co-culture (Figure 2B) and processed via WTA, as described previously [7,8].

Transcriptome analysis revealed peritoneal-seeding roles for cell adhesion molecules such as adenomatous-polyposis coli (APC) (1.8–13.5-fold increase) and vascular endothelial growth factor A (VEGFA) (1.8–11.2-fold increase), as well as growth factors such as insulin growth factor (IGF) (2.1–8,955.7-fold increase; Figure 2C). Notably, MMPs were strongly upregulated to different levels in metastasising tumour cells. MMP2 was upregulated 38,698.4-fold in HT29 and 2.4-fold in Colo205 cells, and MMP9 was upregulated 231.9- and 4727.4-fold, respectively. Similar results were obtained for MMP10 (data not shown). In addition, transcriptome analysis of the ovarian cancer cell line SKOV3 showed similar results for cell adhesion molecules, growth factors, and MMP9 (Appendix A), thereby confirming previous results [16]. These results suggested that MMP2 and MMP9 played pivotal roles during the PC of CRC lines in our ex vivo peritoneum model.

### 3.3. Inhibition of Peritoneal Colonisation of Cancer Cells by MMP2/9 Blockage

MMP2 and MMP9 were strongly upregulated during co-culture with cellularised peritoneal tissue. We therefore investigated the effect of MMP inhibition as a possible treatment for PC. GFP-expressing CRC cells were cultured with human peritoneum and treated with either the pan-MMP inhibitor batimastat or the specific MMP2/9 inhibitor III immediately after seeding onto the peritoneum. As neither batimastat nor the specific MMP2/9 inhibitor III led to any relevant cytotoxicities in our CRC cells, which is in accordance with previous literature [17], inhibitors were used at concentrations that efficiently reduced protein expression levels (data not shown). Treatment with batimastat reduced colonisation of HT29 and Colo205 cells by 36% and 68% (*p* = 0.0073 and *p* = 0.0002), respectively (Figure 3). Similarly, treatment with the specific MMP2/9 inhibitor III reduced peritoneal colonisation of HT29 and Colo205 cells by 50% and 41% (*p* = 0.0003 and *p* = 0.0051), respectively. Functional inhibition of MMP2/9 in SKOV3 cells, as well as combined knockdown of MMP2/9 by siRNA pools, significantly reduced peritoneal colonisation (Appendix A, uncropped WB figures Appendix A), thus indicating active roles for MMP2 and MMP9 activation during the PC of CRC and ovarian cancer cell lines.

We further investigated the molecular mechanisms underlying MMP-enhanced peritoneal attachment and the impaired colonisation of cancer cells after treatment with MMP inhibitors. We first analysed the effect of increased MMP9 expression in patient-derived PC. Active MMPs are known to cleave fibronectin in the ECM, as a prerequisite for the successful invasion and dissemination of cancer cells. We therefore determined if increased MMP9 activity in patient-derived PC samples was correlated with increased fibronectin cleavage. The ratio of cleaved/intact fibronectin increased from 0.19 to 1.74 in healthy tissues to 2.08–10.23 in PC samples (1.21–53.84-fold increase) (Figure 4A). These results suggest that enhanced cleavage of fibronectin might be due to enhanced MMP2/9 activity in PC.

Given that MMP activity correlated with fibronectin cleavage in patient-derived PC samples, we also investigated if the reduced peritoneal colonisation of cancer cells following MMP inhibition was related to reduced fibronectin cleavage. HT29 or SKOV3 cells were seeded onto peritoneum tissue and treated with batimastat or MMP2/9 inhibitor III. Protein was isolated from the whole tissue 48 h after seeding and subjected to immunoblotting. As shown in Figure 4B, treatment of SKOV3 cells, but not HT29 cells, with batimastat or MMP2/9 inhibitor III reduced the ratio of cleaved/intact fibronectin in peritoneal tissue by 49% and 80%, respectively, compared with peritoneal tissue cultured with SKOV3 cells without inhibitor treatment (not significant). MMP inhibition thus reduced the amount of cleaved fibronectin compared with peritoneal tissues cultured with SKOV3 cells alone. If the successful peritoneal colonisation of cancer cells depends on fibronectin cleavage, these results may help to explain why treatment with MMP inhibitors reduced tissue colonisation by cancer cells.

## 4. Discussion

PC progression is a multi-step process that differs mechanistically from organ metastasis. We hypothesised that MMP2 and MMP9, both of which are relevant for ECM degradation, may be necessary for peritoneal invasion and cancer cell colonisation [18,19,20]. MMP2/9 gene expression levels were overexpressed in patient-derived primary CRC and corresponding PC samples. In addition, MMP2 and MMP9 were strongly upregulated following peritoneal colonisation by CRC cell lines in our human ex vivo model, reflecting the phenotype in CRC patients with advanced-/metastasised stage and poor survival. Our results corresponded to findings in ovarian carcinoma cells, in which the upregulation of MMP2 mRNA expression was associated with enhanced peritoneal adhesion [16]. Furthermore, the inhibition of MMP2 activity reduced the attachment of ovarian cancer cells in a 3D organotypic model of metastatic ovarian cancer [16]. However, the current study provides the first evidence for the role of MMP2/9 using a humanised CRC model.

Compared with other well-known factors supporting metastasis (e.g., APC, VEGFA, IGF), MMP2/9 expression was pronounced in adhering to CRC cells on human peritoneum, suggesting an important role for ECM remodelling molecules during this process. Interestingly, we found that MMP genes were strongly upregulated in CRC cell lines after contact with living peritoneal cells (cellularised vs. decellularised peritoneum). MMP overexpression in primary tumours might thus be enhanced upon contact with peritoneal cells, further promoting cancer cell attachment to the peritoneum and subsequent invasion. In addition, cleaved fibronectin, which favours cancer cell invasion, was enhanced in PC samples (*n* = 3) from patients with primary CRC, as well as during peritoneal colonisation by ovarian cancer cells, but not CRC cells, in our human ex vivo peritoneum model. As tumour cells differ in their signalling network due to their underlying mutations, MMP-regulated pathways might differ between SKOV3 and HT29 cells. It is therefore possible that fibronectin cleavage might fully depend on proper MMP2/9 function in SKOV3 cells, while, in HT29 cells, fibronectin cleavage might depend on additional pathways apart from MMP2/9. These results were in accordance with previous results showing that peritoneal adhesion of the ovarian cancer cell line OvCa was caused by MMP2-mediated fibronectin and vitronectin cleavage [16].

Because MMPs played an essential role during PC formation in our model, we speculated that the inhibition of MMP2/9 could serve as an effective therapy to prevent PC at an early stage in patients with CRC [21]. Within the MMP family, both MMP2 and MMP9 belong to the gelatinases and possess overlapping functions [18,19]. Therefore, MMP2 and/or MMP9 might be involved in tumour cell invasion. Indeed, treatment with either a selective MMP2/9 inhibitor or the pan-MMP inhibitor, batimastat, significantly reduced peritoneal colonisation and thus PC formation. This is in line with the results of Wang et al., who showed that downregulation of MMP1 expression inhibited the progression of CRC in vitro and in vivo [22]. Moreover, treatment with MMP1 short hairpin RNA reduced the expression of ECM molecules such as N-cadherin and vimentin and the expression of p-Akt and c-Myc, which are involved in tumour and metastasis formation [22].

Recent clinical trials failed to prevent the development of PC in high-risk CRC patients using conventional hyperthermic intraperitoneal therapy. However, MMP inhibition has not yet been tested in this context. Batimastat is a pan-MMP inhibitor that has shown favourable responses in clinical trials for several tumour entities and malignant ascites with no major toxicities [23,24,25]. Our data support the use of batimastat in a clinical trial in patients at high risk of PC development.

Our human ex vivo model had several limitations. First, we used commercial cancer cell lines for co-culture with peritoneal tissue to mimic PC. This was because the peritoneal tissue was obtained from non-cancer patients, and co-culture with peritoneal or CRC cells from the same patient was therefore not possible. Second, our model did not include blood circulation or immune system components. The model therefore allowed for the investigation of interactions between cancer cells and the tumour microenvironment, but lacked the ability to investigate the effects of additional modifications via the immune system, such as cytokines. To overcome some of these limitations in the future, the model could be based on peritoneum and tumour cells isolated from the same patient during cytoreductive surgery. Moreover, PBMCs isolated from the patients’ blood could enable the inclusion of the immune system. Nevertheless, our ex vivo co-culture model provides an easy and inexpensive model for investigating human tumour cells within a human microenvironment in contrast to animal models.

## 5. Conclusions

Our results highlight the relevance of the MMP2/9 pathway in the development of PC in patients with CRC and suggest a possible therapeutic window in which to prevent PC at the stage of cancer cell invasion.

## Figures and Tables

**Figure 1 cancers-14-03760-f001:**
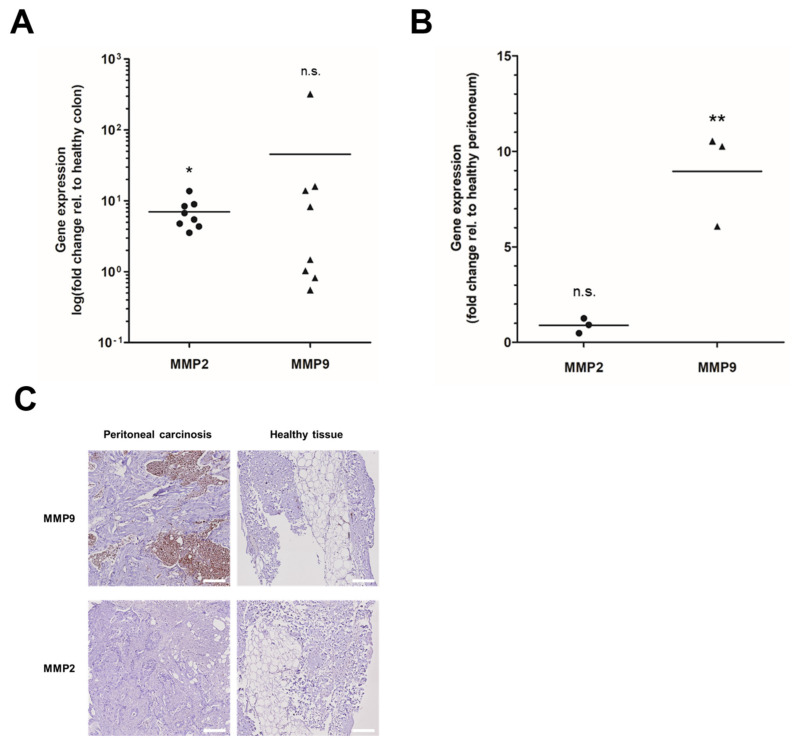
MMPs are overexpressed in CRC and patient-derived PC: (**A**,**B**) RNA was isolated from FFPE sections of (**A**) primary colorectal tumours (*n* = 8) or (**B**) PC (*n* = 3) and adjacent healthy tissue and analysed for MMP2 (dots) and MMP9 (triangles) gene expression using RT-qPCR; two-sided unpaired *t*-test relative to healthy colon/peritoneum; * *p* < 0.05, ** *p* < 0.01, n.s. not significant. (**C**) FFPE sections (3 µm) of PC or adjacent healthy tissue were stained with antibodies against MMP9 or MMP2. The tissue of PC was strongly positive for MMP9 and weakly positive for MMP2. Healthy tissue was negative for MMP9 and MMP2; Scale bar 200 µm.

**Figure 2 cancers-14-03760-f002:**
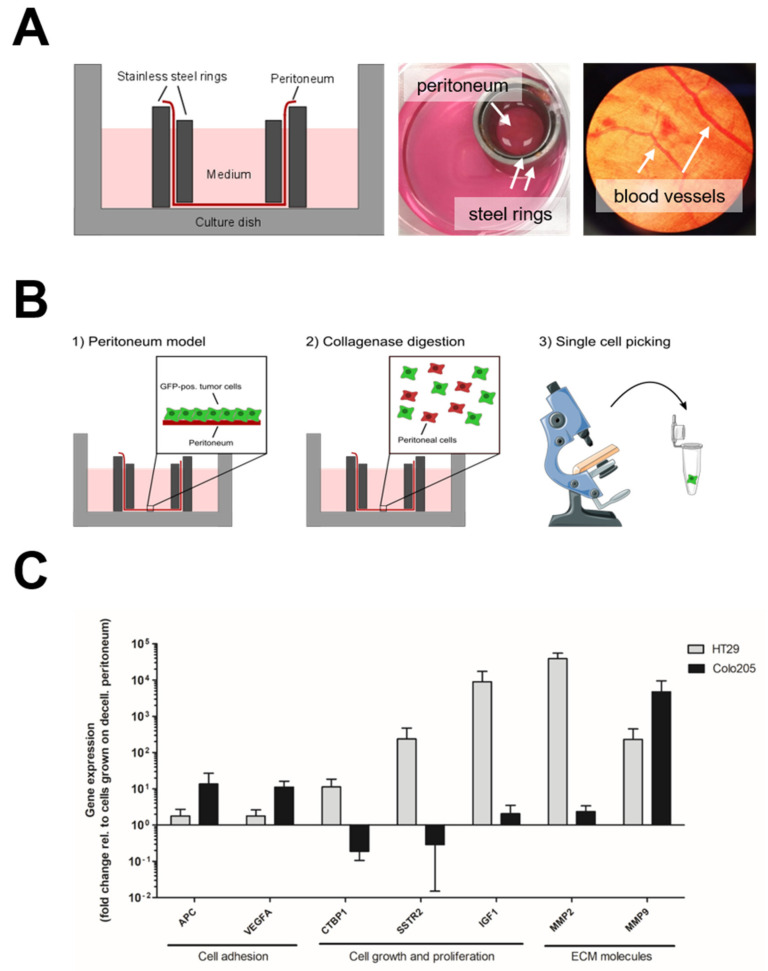
Transcriptome analysis reveals a role for MMPs during peritoneal metastasis of CRC cells: (**A**) Schematic setup of the human ex vivo peritoneal model (left) and photographic images of the peritoneal model (middle, right). (**B**) Scheme of the experimental set-up. GFP-pos. tumour cells were seeded on patient-derived peritoneum. After 24 h colonisation, the tissue was digested with collagenase to release attached tumour cells. Afterwards, GFP-pos. single cells were isolated under a fluorescence microscope. Medical images were obtained from smart.servier.com (accessed on 22 April 2022). (**C**) GFP-pos. HT29 and Colo205 cells were seeded on cellularised (including peritoneal cells) or decellularised (without peritoneal cells, ECM components only) peritoneum and isolated 24 h after colonisation. Afterwards, RNA was isolated from single tumour cells. RNA-expression analysis shows upregulation of several genes associated with cell adhesion, cell growth, or ECM composition when cells were grown on cellularised compared with decellularised peritoneum; Mean and SE of at least three independent experiments.

**Figure 3 cancers-14-03760-f003:**
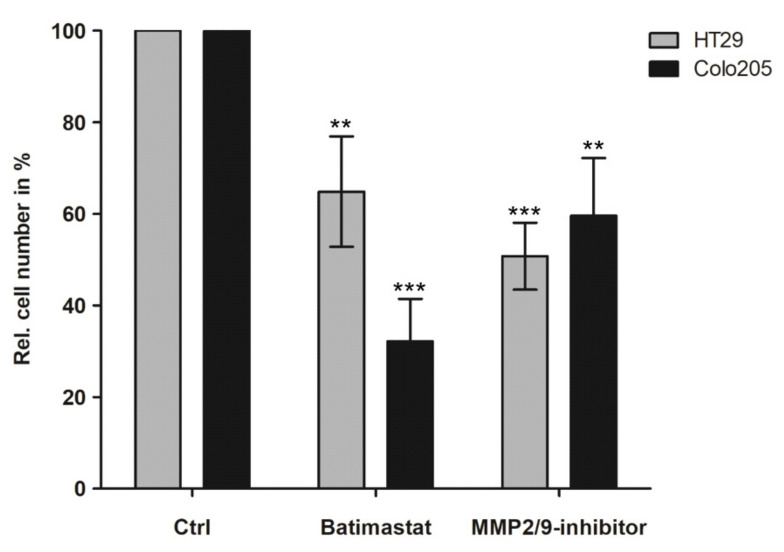
MMP2/9-inhibitors reduce colonisation of CRC cells seeded on the peritoneum: GFP-pos. HT29 and Colo205 cells were seeded on patient-derived peritoneum and simultaneously treated with MMP-inhibitors. After 24 h, cells were released from the peritoneum via collagenase digestion and counted using a Neubauer chamber; Mean and SE of at least three independent experiments; untreated control normalised to 100%; two-sided, unpaired *t*-test ** *p* < 0.01, *** *p* < 0.001.

**Figure 4 cancers-14-03760-f004:**
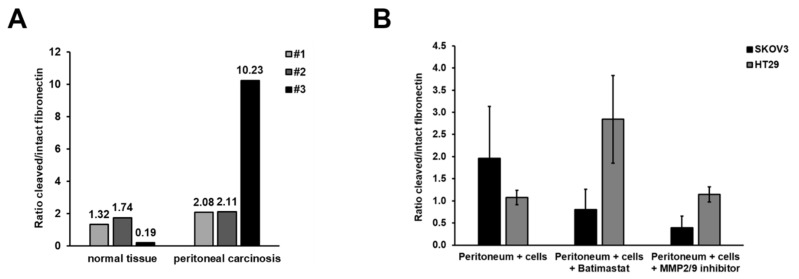
Fibronectin cleavage is enhanced in PC and can be reduced by MMP2/9 inhibition: (**A**) Protein was isolated from patient-derived peritoneum (*n* = 3) with and without PC and subjected to immunoblotting. The signal for fibronectin and cleaved fibronectin was normalised to actin, and the ratio of cleaved vs. intact fibronectin was calculated. Fibronectin cleavage was enhanced in tissues with PC compared with healthy peritoneum (uncropped WB figures Appendix A). (**B**) HT29 or SKOV3 cells were seeded on patient-derived peritoneum and simultaneously treated with MMP-inhibitors (250 µM each). After 48 h, protein was isolated from the whole tissue and subjected to immunoblotting. The signal for fibronectin and cleaved fibronectin was normalised to actin, and the ratio of cleaved vs. intact fibronectin was calculated. Fibronectin cleavage was inhibited by batimastat or MMP2/9-inhibitor III treatment in SKOV3 but not HT29 cells. Mean and SE of at least three independent experiments (uncropped WB figures Appendix A).

## Data Availability

The data presented in this study are available on request from the corresponding author.

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
