# Peer review of "Pharmacologic Targeting of MMP2/9 Decreases Peritoneal Metastasis Formation of Colorectal Cancer in a Human Ex Vivo Peritoneum Culture Model"

_cancers, 2022, doi:10.3390/cancers14153760_

Round 1

Reviewer 1 Report

1)     Figure 2.  Do you have images of the cells growing in the tissue? It would be interesting to understand how the cells grow in this model. Did you evaluate expression overtime or only 24 hours? For colo205 cell proliferation results could be because cell death? Do you apoptosis data?

2)     S1. Did you evaluate expression overtime or only 24 hours?

3)     Did you evaluate the MMP2/9 in the colon cancer cells? Didn’t work?

4) Did you measure the level inhibition level of MMP2/9 of batimastat or MMP2/9 inhibitor III? Which is the IC50 for all the cell lines or the cytotoxicity. Include information of the concentration used for all cell lines and drugs.

I think it is really important to have IC50 if the inhibitors, or cytotoxicity and concentrations 

Author Response

We would like to thank reviewer #1 for his/her thorough review of our article and his/her helpful comments which helped us to produce a stronger manuscript altogether.

1) Figure 2.  Do you have images of the cells growing in the tissue? It would be interesting to understand how the cells grow in this model.

We thank the reviewer for his/her additional comment and we agree that the analysis of the invasion pattern of cancer cells in our model is of great importance. We have already extensively analysed the growing and invasion pattern of cancer cell lines into the peritoneal tissue, which was published previously by Mönch et al, A human ex vivo co-culture model to investigate peritoneal metastasis and innovative treatment options, 2021, Pleura and Peritoneum.

Did you evaluate expression overtime or only 24 hours?

GFP-expressing cells were firmly attached to the tissue 24 h after seeding them on the peritoneum and thus, expression analyses were performed after 24 h.

For Colo205 cell proliferation results could be because of cell death? Do you have apoptosis data?

As Colo205 cells are semi-adherent cells derived from ascites, these cells might behave slightly different when attaching to the peritoneum. However, we thought it would be of special interest to analyse how gene expression of Colo205 changes upon attachment to the peritoneum. We do not have any hint that cells might undergo apoptosis during attachment to the peritoneum but rather proliferate within the tissue (Mönch et al, A human ex vivo co-culture model to investigate peritoneal metastasis and innovative treatment options, 2021, Pleura and Peritoneum).

2) S1. Did you evaluate expression overtime or only 24 hours?

As stated above, GFP-expressing cells were firmly attached to the tissue 24 h after seeding them on the peritoneum and thus, expression analyses were performed after 24 h.

3) Did you evaluate the MMP2/9 in the colon cancer cells? Didn’t work?

It is justified that the reviewer addresses this question and we have extensively analysed MMP2/9 gene and protein expression in CRC cells as well. However, siRNA mediated knockdown did not lead to reduction in MMP2/9 protein levels in CRC ells.

4) Did you measure the level inhibition level of MMP2/9 of batimastat or MMP2/9 inhibitor III? Which is the IC50 for all the cell lines or the cytotoxicity. Include information of the concentration used for all cell lines and drugs. I think it is really important to have IC50 if the inhibitors, or cytotoxicity and concentrations.

We thank the reviewer for this helpful comment and have included an additional explanation in the section “Inhibition of peritoneal colonisation of cancer cells by MMP2/9 blockage” in the results part (lines 269-272). As stated there, we have investigated cell survival upon administration of batimastat or MMP2/9 inhibitor III. However, we did not see any relevant cytotoxicities in our cells, which is in accordance with previous literature (Botos et al, Batimastat a potent mmp inhibitor exhibits an unexpected mode of binding, 1996, Biochemistry). Therefore, inhibitors were used at concentrations that efficiently reduced protein expression levels in our cells (data not shown).

Reviewer 2 Report

thank you for giving me the opportunity to review this very interesting paper exploring the complex mechanisms behind peritoneal carcinosis. the methodology is clearly expressed as well as the results. the synthesis of the literature allows us to understand the clinical problem of therapeutic management. however, some questions remain regarding the interpretation of the results.

« In patient-derived PC samples, mRNA levels of MMP9, but not MMP2, were increased 6–10.5- 210 fold (p=0.0053) compared with adjacent healthy peritoneum (n=3; Fig 1B). Immunohisto-chemistry staining accordingly revealed strong expression of MMP9, but not MMP2, in  PC compared with healthy tissues (Fig. 1C). » how the authors explain these conflicting results ?

« As shown in Fig. 4B, treatment of SKOV3 cells, but not HT29 cells, with batimastat or MMP2/9 inhibitor III reduced the ratio of cleaved/intact fibronectin in peritoneal tissue by 49% and 80%, respectively, compared with peritoneal tissue cultured with SKOV3 cells without inhibitor treatment (not significant). MMP inhibition thus reduced the amount of 314 cleaved fibronectin compared with peritoneal tissues cultured with SKOV3 cells alone. » how the authors explain these conflicting results ?

. how do the authors intend to use other models of peritoneal carcinosis in the future to overcome the limitations of their model?

Author Response

We would like to thank reviewer #2 for his or her additional comments which helped us to produce a stronger manuscript altogether.

1) In patient-derived PC samples, mRNA levels of MMP9, but not MMP2, were increased 6–10.5- fold (p=0.0053) compared with adjacent healthy peritoneum (n=3; Fig 1B). Immunohistochemistry staining accordingly revealed strong expression of MMP9, but not MMP2, in PC compared with healthy tissues (Fig. 1C). How the authors explain these conflicting results?

The reviewer addresses a very good point asking why only MMP9 but not MMP2 was upregulated in patient-derived PC samples. We have included an additional paragraph in the discussion section (lines 366-368) stating that within the MMP family both, MMP2 and MMP9, belong to the gelatinases and have overlapping functions. Thus, tumour cell invasion might already be achievable via upregulation of either MMP2 or MMP9.

2) As shown in Fig. 4B, treatment of SKOV3 cells, but not HT29 cells, with batimastat or MMP2/9 inhibitor III reduced the ratio of cleaved/intact fibronectin in peritoneal tissue by 49% and 80%, respectively, compared with peritoneal tissue cultured with SKOV3 cells without inhibitor treatment (not significant). MMP inhibition thus reduced the amount of cleaved fibronectin compared with peritoneal tissues cultured with SKOV3 cells alone. How the authors explain these conflicting results?

We thank the reviewer for this helpful comment and we totally agree with the reviewer that the results in Fig. 4B obtained with SKOV3 and HT29 cells are contradictory. We have included an additional paragraph in the discussion (lines 355-362) stating that tumour cells differ in their signalling network due to their underlying mutations, and thus MMP regulated pathways might differ between SKOV3 and HT29 cells. It is therefore possible that in SKOV3 cells, fibronectin cleavage might fully depend on proper MMP2/9 function, while in HT29 cells fibronectin cleavage might depend on other pathways apart from MMP2/9.

3) How do the authors intend to use other models of peritoneal carcinosis in the future to overcome the limitations of their model?

We appreciate the reviewers very good comment and have included an additional paragraph in the discussion section (lines 388-393). In the future, we will also focus on cultivation of peritoneum from cancer patients using co-cultures with cancer cells and peritoneum from the same patient. This might also offer the possibility to add immune cells (PBMCs) to the system to study immune cell- tumour cell interactions.

Round 2

Reviewer 1 Report

All the comments were answered, I would  suggest to add more information about the ex vivo model, pictures, etc.

Author Response

We would like to thank reviewer #1 again for his/her helpful comments which has helped us to produce a better manuscript.

1) I would suggest to add more information about the ex vivo model, pictures, etc.

We are especially thankful for the reviewers suggestion and have included an additional description of the model in the results section (lines 225-232) as well as an additional subfigure in Figure 2. Here, we show the schematic set-up of the model as well as photographic images for better illustration.